# A 2D-GRAPPA Algorithm with a Boomerang Kernel for 3D MRI Data Accelerated along Two Phase-Encoding Directions

**DOI:** 10.3390/s23010093

**Published:** 2022-12-22

**Authors:** Seonyeong Shin, Yeji Han, Jun-Young Chung

**Affiliations:** 1Department of Neuroscience, College of Medicine, Gachon University, Incheon 21988, Republic of Korea; 2Department of Health Sciences and Technology, GAIHST, Gachon University, Incheon 21988, Republic of Korea; 3Department of Biomedical Engineering, Gachon University, Seongnam 13120, Republic of Korea

**Keywords:** 3D MRI, acceleration in two phase-encoding directions, 2D-GRAPPA algorithm, shapes and dimensions of a kernel, boomerang kernel

## Abstract

For the reconstruction of 3D MRI data that are accelerated along the two phase-encoding directions, the 2D-generalized autocalibrating partially parallel acquisitions (GRAPPA) algorithm can be used to estimate the missing data in the k-space. We propose a new boomerang-shaped kernel based on theoretic and systemic analyses of the shape and dimensions of the kernel. The reconstruction efficiency of the 2D-GRAPPA algorithm with the proposed boomerang-shaped kernel (i.e., boomerang kernel (BK)-2D-GRAPPA) was compared with other 2D-GRAPPA algorithms that utilize different types of kernels (i.e., EX-2D-GRAPPA and SK-2D-GRAPPA) based on computer simulation, phantom and in vivo experiments. The proposed method was validated for different sets of ACS lines with acceleration factors from four to eight and various sizes of the kernels. A quantitative analysis was also performed by comparing the normalized root mean squared error (nRMSE) in the images and the undersampled edges. Computer simulation, in vivo and phantom experiments, and the quantitative analysis, showed that the proposed method could reduce aliasing artifacts without reducing the SNRs of the reconstructed images.

## 1. Introduction

Data acquisition time and image quality are two of the most important considerations in clinical MRI. A long acquisition time causes patient discomfort and degrades the image quality due to voluntary or involuntary movements (e.g., blood flow, eye movement, respiration, cardiac motion, etc.). To reduce acquisition time while maintaining image quality, undersampling is commonly used in combination with phased array RF coils. In parallel MR imaging, MR signals are simultaneously collected from the multi-channel receive RF coils and the acquired signals are combined from a multitude of closely positioned MR receive coils. Thus, many parallel imaging (PI) techniques have been developed to estimate the unacquired MR signals and to reconstruct images from the undersampled data using multi-channel coil information. To this end, researchers have developed advanced reconstruction methods, which collect a portion of k-space data and estimate unacquired or missing data points, including sensitivity encoding (SENSE), simultaneous acquisition of spatial harmonics (SMASH), AUTO-SMASH and generalized autocalibrating partially parallel acquisitions (GRAPPA) [1,2,3,4,5,6,7,8,9]. Other advanced methods include SAKE [10], AC-LORAKs [11], PRUNO [12] and HICU [13], which rely on the linear predictability in the MRI data [14]. In these algorithms, the unfolded image or missing data points can be reconstructed from sensitivity maps [1], from a linear combination of neighboring data points [7,8,14,15], by utilizing a structured low-rank matrix [10,11,16,17] or a convolutional framework [13].

According to the nature of combining multi-channel images, PI techniques can be categorized as image-based or k-space-based reconstruction algorithms. SENSE [6] is one of the most widely used image-based PI techniques that compensate for aliasing artifacts in the image domain. Other algorithms, such as SMASH [1], AUTO-SMASH [2] and GRAPPA [7], estimate the unacquired k-space signals in the k-space domain. For example, the sensitivity profile of each channel is used to generate an interpolation kernel and the combined full k-space data is estimated from the undersampled multi-channel k-space data. In the advanced version of SMASH (AUTO-SMASH), the interpolation kernel is generated with a self-calibrating scheme using additional k-space signals. In GRAPPA, autocalibrating signal (ACS) lines are used to generate an interpolation kernel for each RF coil channel and to estimate the full k-space data for each channel. The generated k-space data are used to reconstruct images for each channel and the final image is reconstructed by combining the multi-channel images. In addition to these conventional reconstruction methods, a number of machine-learning-based approaches have been recently proposed where deep-learning networks are used for MR image reconstruction [18,19,20]. However, sensitivity encoding (SENSE) or generalized autocalibrating partially parallel acquisitions (GRAPPA) are the most frequently used approaches as they are straightforward to implement.

In terms of MR acquisition, images can be acquired using 2D or 3D pulse sequences. In a 2D sequence, each radiofrequency (RF) pulse excites a narrow slice and the signals are collected in a single k-space, which can be transformed into a single 2D image. In a 3D sequence, each RF pulse excites the entire imaging volume (i.e., multiple image slices) and phase-encodings are performed in two directions to discriminate between different slices. Thus, in the case of 3D imaging, it requires an even longer imaging time than 2D imaging. While the previously mentioned PI techniques were initially proposed to reconstruct images acquired with 2D pulse sequences, the acquisition time of 3D pulse sequences can also benefit from 2D parallel imaging (PI) techniques, where undersampling is performed in k-space along the two PE directions. To reconstruct the 3D MRI data that are accelerated with 2D PI, several algorithms have been proposed. For example, some of the SENSE-based reconstruction techniques utilize the 1D SENSE algorithm [21] or the optimized sampling pattern [22]. However, SENSE-based techniques may result in increased image artifacts because patient motion could generate inaccurate coil sensitivity maps [23]. On the other hand, GRAPPA-based algorithms, such as 2D-GRAPPA and 2D-GRAPPA-Operator (OP), do not require sensitivity maps and can directly reconstruct images from undersampled data. Instead of using sensitivity maps, GRAPPA estimates unacquired k-space data points by a linear combination of the acquired data points, and a kernel specifies which surrounding data points are used for reconstruction. Once the shape of the kernel is determined, the weighting coefficients are computed on the autocalibration signal (ACS) lines and the generated kernel (i.e., set of weighting values) then reconstructs the missing data through the convolution process. Since the calculated weighting values are applied in all regions, selection of the kernel has a significant impact on image quality, as well as on reconstruction time [24].

In this paper, we propose a new 2D-GRAPPA algorithm with a boomerang-shaped kernel (BK-2D-GRAPPA) based on a theoretical and systematic analysis of the existing kernels. The proposed boomerang kernel is defined by the entirety of the acquired data points utilized with the kernels of the EX-2D-GRAPPA algorithm. The kernel is designed to exploit more spatial information compared to other 2D-GRAPPA algorithms and allocates less memory for implementation. To evaluate the efficiency of the boomerang kernel, the proposed method was analyzed with respect to noise, the amount of k-space reference data, the total acceleration factor (AF) and the size of the kernels. In addition, a quantitative evaluation was performed by comparing the mean of the normalized root mean square error (nRMSE) of different channel images.

## 2. Related Works

As mentioned in the introduction section, the GRAPPA-based reconstruction algorithms utilize kernels to determine which acquired data points are used for estimation of the missing (or unacquired) k-space data. In other words, the kernel shape defines which surrounding data points are used for estimation, thereby affecting general image quality. In addition to the kernel shapes, the step-by-step procedure of estimation or how the kernels are applied during the reconstruction is also important for image quality. For reconstruction of the 3D dataset undersampled in two dimensions, two GRAPPA-based approaches can be mainly considered. The first approach is called 2D-GRAPPA-OP, which splits the reconstruction of the 3D dataset into two separate consecutive 1D-GRAPPA reconstructions, each performed along one direction [25]: estimating one phase-encoding line (*k_y_*) followed by another (*k_z_*). As reconstruction is performed in a consecutive manner, the estimated data from the precedent phase-encoding direction (*k_y_*) are used to estimate the unacquired data in the other phase-encoding direction (*k_z_*). In other words, 2D-GRAPPA-OP may show a propagation of errors because it uses preceding reconstructed missing data in the second data reconstruction step along another phase-encoding direction. Additional reconstruction errors may be introduced due to the imperfect reconstructions in the first reconstruction step [25,26].

Another way to reconstruct a 3D dataset that has been undersampled in two dimensions is to use 2D-GRAPPA, which uses an extended GRAPPA reconstruction. Unlike 2D-GRAPPA-OP, 2D-GRAPPA reconstructs missing data points only using measured points. For 2D-GRAPPA, several types of kernels were proposed [27,28,29]. The simplest method for reconstructing undersampled 3D data is to separately use a 1D kernel in both the *k_y_* and *k_z_* directions and a 2D kernel in the *k_y_*-*k_z_* plane [28]. The extended kernel (EX)-2D-GRAPPA algorithm uses three types of 2D kernels in the *k_y_*-*k_z_* plane, two of which are expanded from 1D kernels along each PE direction in order to enhance the signal-to-noise ratio (SNR). The single-kernel (SK)-2D-GRAPPA algorithm utilizes a single square kernel, which is defined by the acquired data points commonly used with the kernels of the EX-2D-GRAPPA algorithm [29], to concurrently reconstruct *R* − 1 missing data points. The SK-2D-GRPPA algorithm requires less memory allocation and fewer arithmetic operations than the EX-2D-GRAPPA algorithm, but suffers from increased aliasing artifacts. The details of the kernels will be discussed in the following sections.

## 3. Materials and Methods

### 3.1. Methods

#### 3.1.1. 2D-GRAPPA Algorithm and Its Basic Kernels

Three-dimensional MRI data, accelerated in two PE directions, can be reconstructed with the 2D-GRAPPA algorithm [26]. The 2D-GRAPPA algorithm reconstructs a missing data point in k-space with a linear combination of the acquired surrounding data points from all coils. A schematic diagram of the 2D-GRAPPA algorithm is shown in Figure 1 and the mathematical expression can be written as follows:(1)$$S_{i}\left(k_{x}\;+\;r_{x}\Delta k_{x},k_{y}\;+\;r_{y}\Delta k_{y},k_{z}\;+\;r_{z}\Delta k_{z}\right)\;\;=\sum_{j=1}^{J}\sum_{\left(s,p,q\right)\in \Omega }W_{i,r_{x},r_{y},r_{z}}\left(j,s,p,q\right)S_{j}\left(k_{x}\;+\;s\Delta k_{x},k_{y}\;+\;pR_{y}\Delta k_{y},k_{z}\;+\;qR_{z}\Delta k_{z}\right),\;$$
where *S* represents a signal, and *i* denotes an individual coil channel of a *J*-element coil array. *r_x_*, *r_y_* and *r_z_* are relative offsets to the position of a missing data point. ∆*k_x_*, ∆*k_y_* and ∆*k_z_* are the sampling intervals in the *k_x_*, *k_y_* and *k_z_* directions, respectively. Ω is a set of indices representing the position of the acquired data points. *R_y_* and *R_z_* are the AFs along the *k_y_* and *k_z_* directions, respectively. *W_i_,_rx_,_ry_,_rz_*(*j,s,p,q*) is a weighting value.

More specifically, the 2D-GRAPPA algorithm can be divided into two processing steps (Figure 1): (i) the calibration of weighting values and (ii) the synthesis of missing data points. When the values of S*_i_* and S*_j_* are known in Equation (1), the weighting values can be determined by solving the inverse problem. Thus, calibration is performed on the reference data obtained without undersampling and can be written as follows [30]:(2)$$S_{tgt}=S_{src}W,$$
where **S***_tgt_* and **S***_src_* are matrices consisting of target and source data, which are extracted from the k-space reference data and have the same configuration as in the kernel’s reconstruction of the missing data in the second step. **W** is a weighting matrix. The inverse problem of Equation (2) can be expressed as follows:(3)$$W={\left(\left(S_{src}^{H}S_{src}\right)\right)}^{-1}\left(S_{src}^{H}S_{tgt}\right)$$

*^H^* represents the conjugate transpose.

In the calibration step, the size of matrix **S** can be increased to calculate more accurate weighting values [3]. For each possible location in the k-space where the weights could be estimated, the target data are arranged into a column of **S***_tgt_*, and the source data are arranged into a row of **S***_src_*. If *n_src_* acquired data points are used for one target point and *N_f_* is the total number of locations considered, **S***_tgt_* and **S***_src_* have [*N_f_* × *J*] and [*N_f_* × *n_src_J*] dimensions, respectively. The dimension of **W** is [*n_src_J* × *J*]. In actual implementation conditions, 3*N_f_n_src_J* + *N_f_J* memory allocations are required to create matrices. (*n_src_J*)^2^(2*N_f_* − 1) + *n_src_J*^2^(2*N_f_* − 1) + *n_src_J*^2^(2*n_src_J* − 1) arithmetic operations are needed to calculate weighting values.

A missing k-space data point is reconstructed with the multiplication of the acquired surrounding data points and precalculated weighting values **W**. To achieve fully reconstructed k-space data, **W** is slid across the entirety of the k-space. The synthesis step can be written as follows:(4)$${\hat{S}}_{rec}=S_{acq}W,$$where **S***_rec_* is a matrix of reconstructed data points, and **S***_acq_* is a matrix of acquired data points. If *N_rec_* is the total number of missing data points to be reconstructed in 3D k-space data, **S***_acq_* has [*N_rec_* × *n_src_J*] dimensions. Therefore, *N_rec_n_src_J* + *n_src_J*^2^ memory allocations and *N_rec_J*(2*n_src_J* − 1) arithmetic operations are required in the synthesis step.

Various types of kernels (or “sets of weights”) can be utilized to reconstruct accelerated 3D k-space data. When a missing data point is located in the same *k_y_* (or *k_z_*) position as the acquired data (first two figures in the first and second rows in Figure 2b), either a 1D kernel that uses acquired data points in the *k_y_* (or *k_z_*) direction or a 2D kernel that uses acquired data points in the *k_y_-k_z_* plane can be utilized. When the missing k-space data point is not located in the same *k_y_* and *k_z_* positions as the acquired data (third figure in the second row in Figure 2b), it can be reconstructed with a 2D kernel using acquired data points in the *k_y_-k_z_* plane. A kernel can also be extended along the *k_x_* direction to improve the quality of reconstruction [8]. The acquired data points can be symmetrically or asymmetrically utilized from a missing data point. In this study, symmetrically acquired neighboring points are assumed.

In Section 3.1.2, Section 3.1.3 and Section 3.1.4, 2D-GRAPPA algorithms using various types of kernels will be introduced and the proposed boomerang kernel will be explained in Section 3.1.5. To simplify the concepts, we would eliminate the *k_x_* dimension, i.e., *r_x_* and *s* = 0 in Equation (1).

#### 3.1.2. The Lowest-Dimensional-Kernel (LK)-2D-GRAPPA Algorithm

The lowest-dimensional (LK)-kernel-2D-GRAPPA algorithm uses three different kernels: two 1D kernels utilizing the acquired data points along each of the two PE directions (i.e., *k_y_* and *k_z_*) and a 2D kernel utilizing the acquired data points in the *k_y_*-*k_z_* plane [23]. The basis kernels of the LK-2D-GRAPPA algorithm are shown in Figure 3a. Since each kernel independently operates to reconstruct the missing k-space data, the number of memory allocations and arithmetic operations required for reconstruction is equivalent to the total number of elements from all three kernels. Table 1 shows a numerical example of the memory allocations and arithmetic operations in each step with the basis kernels when the matrix size is 100 × 62 × 62 (# of readout × # of phase encoding × # of partition encoding), with a total AF of four (*R_y_* × *R_z_* = 2 × 2) and 100 × 24 × 24 k-space reference data.

#### 3.1.3. The Extended-Kernel (EX)-2D-GRAPPA Algorithm

The 1D kernels of the LK-2D-GRAPPA algorithm can be extended in the orthogonal PE direction to consider the correlation between the missing data points and the surrounding acquired points [8]. The algorithm that uses three types of 2D kernels in the *k_y_*-*k_z_* plane is called the extended-kernel (EX)-2D-GRAPPA algorithm [23]. The basis kernels of the EX-2D-GRAPPA algorithm are shown in Figure 3b. As with the LK-2D-GRAPPA algorithm, each kernel independently operates to reconstruct the missing k-space data. The number of memory allocations and arithmetic operations of the EX-2D-GRAPPA algorithm is also described in Table 1. The EX-2D-GRAPPA algorithm needs a greater number of memory allocations and arithmetic operations than the LK-2D-GRAPPA algorithm.

#### 3.1.4. The Single-Kernel (SK)-2D-GRAPPA Algorithm

In the single-kernel (SK)-2D-GRAPPA algorithm, *R_y_R_z_*–1 missing data points are simultaneously reconstructed with a single square kernel [24]. Kernels in [20,21] could be considered as an extension of the basis kernel of the SK-2D-GRAPPA algorithm. The process of composing a square kernel is shown in Figure 3c. Given that the kernels of the EX-2D-GRAPPA algorithm are arranged to reconstruct a block, a kernel that uses the common acquired data points utilized with the kernels of the EX-2D-GRAPPA algorithm is defined as a square kernel. One of the four types of square kernels can be generated depending on the definition of a block.

In the calibration step, the number of locations considered to estimate a kernel, *N_f_*, differs by the types of kernels in terms of factors such as the amount of reference data or the acquisition scheme. In our experimental conditions, a type-4 single kernel is utilized to consider more information for the calculation of weighting values compared to the other types of single kernel. The number of memory allocations and arithmetic operations of the SK-2D-GRAPPA algorithm is shown in Table 1. The SK-2D-GRAPPA algorithm requires a smaller number of memory allocations and arithmetic operations than the EX-2D-GRAPPA algorithm. However, the SK-2D-GRAPPA algorithm shows more residual aliasing artifacts than the EX-2D-GRAPPA algorithm due to the reduced number of acquired data points that are utilized for reconstruction [22].

#### 3.1.5. The Proposed Boomerang Kernel

We propose a boomerang-shaped kernel, which could maximize the use of acquired signals during the reconstruction process, thereby increasing the quality of reconstructed images. The process of composing a boomerang kernel is shown in Figure 3d. Given that the kernels of the EX-2D-GRAPPA algorithm are arranged to reconstruct a block, we define a boomerang kernel that uses the entirety of the acquired data points utilized with the kernels of the EX-2D-GRAPPA algorithm and simultaneously reconstructs *R_y_R_z_* − 1 missing data points. As with the square kernel in the SK-2D-GRAPPA algorithm, one of the four types of boomerang kernel is constructed according to the definition of a block. In our experimental conditions, a type-4 boomerang kernel is created to consider more combinations of ACS lines than other types of boomerang kernels. The required number of memory allocations and arithmetic operations is shown in Table 1. The 2D-GRAPPA algorithm using the boomerang kernel (i.e., BK-2D-GRAPPA algorithm) requires the largest number of arithmetic operations. However, the number of memory allocations is the second smallest because *RyRz* − 1 missing data points of a block are reconstructed at once. Since the boomerang kernel utilizes more acquired data points in the calibration step than other 2D-GRAPPA algorithms, the BK-2D-GRAPPA algorithm can provide reduced residual aliasing artifacts by obtaining more accurate weighting values.

### 3.2. Computer Simulations

The errors generated during the calculation of the weighting values can cause residual aliasing artifacts. To analyze the effects of the boomerang kernel in resolving aliasing artifacts, computer simulations were performed as follows. Simulation datasets were calculated assuming a 12-channel head coil consisting of four clusters, as described in Figure 4. The volume-of-interest of the coil was an isotropic cube with a side length of 28 cm. Each cluster consisted of three circular loop coil elements with a diameter of 7.2 cm, and the circular loops in a cluster were positioned 1 mm apart. Using the Biot–Savart law, a coil sensitivity map was calculated with the following parameters: FOV = 192 × 192 mm^2^, matrix size (# of frequency encoding × # of phase encoding × # of partition encoding) = 192 × 192 × 192 and slice thickness = 1 mm. In addition, to evaluate the efficiency of the boomerang kernel in the presence of noise, a sensitivity map with a 30 dB signal-to-noise ratio (SNR) was also generated by adding white Gaussian noise to the originally generated coil sensitivity map.

Undersampling was performed along the *k_y_* and *k_z_* directions so that datasets with AFs of four (*R_y_* × *R_z_* = 2 × 2) and eight (*R_y_* × *R_z_* = 2 × 4 and 4 × 2) were produced. The simulation datasets were then reconstructed with different 2D-GRAPPA algorithms (i.e., LK-2D-GRAPPA, EX-2D-GRAPPA, SK-2D-GRAPPA and BK-2D-GRAPPA) and compared. Different numbers of k-space reference data were also utilized, i.e., 24 × 24, 32 × 32 and × 48 × 48 (# of reference lines in *k_y_* direction × # of reference lines in *k_z_* direction), to observe the accuracy of data estimation with respect to the number of reference lines. After estimating all missing data, inverse Fourier transform was applied to generate an individual coil image. Two sets of images were reconstructed for each algorithm by including and excluding the k-space reference lines before applying the inverse Fourier transform, thereby investigating the effects of reference data for reconstruction. The final images produced by excluding the reference lines only showed the effects of different kernels.

For each kernel, the basis kernels were selected as follows: (a) LK-2D-GRAPPA: *n_x_* × *n_y_* × *n_z_* = 1 × 1 × 2, 1 × 2 × 1, 1 × 2 × 2 (b) EX-2D-GRAPPA: 1 × 2 × 3, 1 × 3 × 2, 1 × 2 × 2 (c) SK-2D-GRAPPA: 1 × 2 × 2 (type-4 single kernel) (d) BK-2D-GRAPPA: boomerang 1 × 3 × 3 (type-4 boomerang kernel). The kernels were also expanded to utilize three acquired data points in the *k_x_* direction to compare the reconstructed images when each kernel utilizes spatial information in all directions (i.e., expanded kernels, (a) LK-2D-GRAPPA: 3 × 1 × 2, 3 × 2 × 1, 3 × 2 × 2; (b) EX-2D-GRAPPA: 3 × 2 × 3, 3 × 3 × 2, 3 × 2 × 2; (c) SK-2D-GRAPPA: 3 × 2 × 2; (d) BK-2D-GRAPPA: boomerang 3 × 3 × 3).

### 3.3. MRI Data Acquisition

We conducted studies with a 3D anthropomorphic head phantom and in vivo data. The 3D anthropomorphic head phantom and in vivo MRI datasets were obtained using a spoiled gradient-echo sequence in a 3T MR scanner (Siemens Magnetom Verio, Erlagen, Germany) with a 12-channel head coil. The sequence parameters were as follows: TR = 20 ms, TE = 5 ms, flip angle = 25°, FOV = 256 × 256 mm^2^, slice thickness = 1 mm and matrix size (# of readout × # of phase encoding × # of partition encoding) = 256 × 256 × 208 for phantom and 256 × 256 × 224 for in vivo experiment. The fully sampled datasets were retrospectively subsampled along the phase-encoding (*y*) and partition-encoding (*x*) directions. One healthy volunteer signed the informed consent and the datasets were acquired during a study approved by the Institutional Medical Ethics Committee and Review Board (IRB).

As for the computer simulations, two separate experiments (AF = 4 and 8) were performed without any regularization. All reconstruction processes were implemented in MATLAB R2016a (The Mathworks Inc., Natick, MA, USA) with an Intel Core i7 4790 CPU (3.60 GHz) and 24 GB RAM.

### 3.4. Quantitative Analyses

Quantitative analyses were performed by comparing the mean of the normalized root mean squared error (nRMSE) of different channel images. The nRMSE was calculated for each entire channel image and its undersampled edges as follows:(5)$$nRMSE=\sqrt{\frac{\sum_{i=1}^{M}{\left|I_{i}^{Full}\;-\;I_{i}^{Recon}\right|}^{2}}{\sum_{i=1}^{M}{\left|I_{i}^{Full}\right|}^{2}}},\;$$where *M* is the number of voxels in the entire channel image or undersampled edges. *I*_i_*^Full^* is a reference channel image with fully sampled k-space data, and *I*_i_*^Recon^* is a reconstructed channel image. An example of an undersampled edge image can be found at the top of Figure 5. The edges were identified with the Laplacian of Gaussian filtering [31] and undersampled along the two PE directions.

## 4. Results

Images reconstructed from the simulated dataset with AF = 4 (Ry × Rz = 2 × 2) by different 2D-GRAPPA algorithms are presented in Figure 5. The images in Figure 5a,b were reconstructed from noise-free simulation data and the images in Figure 5c,d were reconstructed from noise-added simulation data. The BK-2D-GRAPPA algorithm showed less residual aliasing artifacts than any other 2D-GRAPPA algorithm, regardless of the presence of noise. The amount of residual aliasing artifacts decreased when the kernel was expanded. When the k-space reference data were included at the end of the reconstruction (for inverse Fourier transform), ringing and edge blurring artifacts were observed in the reconstructed images, but the reconstructed images of the BK-2D-GRAPPA algorithm showed minimized ringing and edge blurring artifacts.

Figure 6 shows images reconstructed from the acquired phantom (a,b) and human brain (c,d) data. The reference image reconstructed from fully sampled k-space data, an image showing the pattern of aliasing artifacts and an undersampled edge mask are also presented on the left. The missing k-space data were estimated from acquired data with 24 × 24 k-space reference lines to generate the final images, and the difference images were also calculated by subtracting the reconstructed images from the reference image. In Figure 6a,c, the images show the reconstruction results obtained when the k-space reference lines were excluded before reconstruction. On the other hand, Figure 6b,d shows images obtained when the reference lines were included for reconstruction. Images from the first to the fourth columns were reconstructed with the basis kernels, and images from the fifth to eighth columns were reconstructed with the expanded kernels using the 2D-GRAPPA algorithms named below. In the reconstructed images of the acquired phantom data, the BK-2D-GRAPPA algorithm eliminated more aliasing artifacts than other 2D-GRAPPA algorithms. It was difficult to visually observe significant differences between different algorithms except for the LK-2D-GRAPPA algorithm, which showed severe artifacts in the reconstructed images of in vivo data. Nevertheless, the BK-2D-GRAPPA algorithm showed fewer errors when compared with the reference image than other 2D-GRAPPA algorithms.

Figure 7 and Figure 8 present the reconstruction results of the 2D-GRAPPA algorithms when the total AF was increased to eight (Ry × Rz = 2 × 4 or 4 × 2) for computer simulation and the acquired data, respectively. Figure 7a–d shows the images reconstructed from noise-free and noise-added simulation data, respectively. Figure 7a,c was obtained when the kz direction was more accelerated (i.e., Ry × Rz = 2 × 4), and (b,d) were acquired when the ky direction was more accelerated (i.e., Ry × Rz = 4 × 2). On the left of each figure, a reference image with fully sampled k-space data, an image displaying the pattern of aliasing artifacts and an undersampled edge mask are also presented. In Figure 7a–d, the top rows display the reconstructed images using 32 × 32 k-space reference lines, and the bottom rows show the difference images between the complex-valued reference and the reconstructed images. The images from columns 1–4 and 5–8 were reconstructed with the basis and expanded kernels, respectively. The applied reconstruction algorithms are listed below the difference images.

Figure 8 shows the reconstructed images of the acquired (a,b) phantom and (c,d) human brain data, where asymmetric acceleration factors were used. For (a) and (c), the kz direction was more accelerated (i.e., Ry × Rz = 2 × 4), and for (b) and (d), the ky direction was more accelerated (i.e., Ry × Rz = 4 × 2). In addition to the reconstructed images, a reference image with fully sampled k-space data, an image displaying the pattern of aliasing artifacts and an undersampled edge mask are presented on the left. In Figure 8a–d, the top rows display the reconstructed images using 32 × 32 k-space reference lines, and the bottom rows show the difference images between the complex-valued reference and the reconstructed images. As with the computer simulation, images in columns 1–4 and 5–8 were reconstructed using the basis kernels and the expanded kernels, respectively.

As demonstrated by Figure 5, Figure 6, Figure 7 and Figure 8, the number of aliasing artifacts increased as the total AF increased from 4 to 8. In general, the images reconstructed by the BK-2D-GRAPPA algorithm showed the smallest difference from the reference image in the noise-free (Figure 7a,b) and the noise-added (SNR = 30 dB) simulation data (Figure 7c,d). In addition, the quality of the reconstructed image was improved when the expanded kernel size was utilized. As shown in Figure 8, the BK-2D-GRAPPA algorithm also showed the smallest number of aliasing artifacts in the phantom and in vivo data, regardless of which direction was more accelerated. The number of residual aliasing artifacts also decreased with the expanded kernel size.

Figure 9 displays the nRMSEs of the 2D-GRAPPA algorithms for the noise-free simulation, noise-added (SNR = 30 dB) simulation, acquired phantom and acquired in vivo data, respectively. For a fair comparison, k-space reference data were commonly excluded at the end of the reconstruction. In Figure 9a–d, the first column shows the nRMSEs calculated in the entire image and the second column shows the values calculated in its undersampled edges. The performance trend with regard to the nRMSEs calculated in the entire images was analogous to that for nRMSEs measured in the undersampled edges. Specifically, the nRMSE decreased as the utilized number of k-space reference lines increased. The nRMSE value also decreased when the expanded kernels were used and increased with increased AF. In general, the nRMSE of the BK-2D-GRAPPA algorithm showed the lowest values, followed by the EX-2D-GRAPPA, SK-2D-GRAPPA and LK-2D-GRAPPA algorithms.

## 5. Discussion

A novel kernel for 2D-GRAPPA, called a boomerang kernel, is proposed in this work for use in parallel 3D MR imaging, where acceleration is performed in two phase-encoding directions. The boomerang kernel was analyzed in computer simulation, phantom and in vivo data and the performance of image reconstruction was compared with that of existing kernels. In the first experiment with computer simulation, we presented reconstructed images where the k-space reference lines were either included or excluded at the end of the reconstruction. When the k-space reference lines were included at the end of the reconstruction, ringing and edge blurring artifacts appeared in the reconstructed images. This could be attributed to the discontinuities between the k-space reference data and the reconstructed points [32,33]. Nonetheless, it could be concluded that the BK-2D-GRAPPA algorithm reconstructed images from the accelerated k-space data, exhibiting the smallest differences from the reference image, as presented in Figure 5.

In the 2D-GRAPPA algorithms, errors may arise from model imperfections and noise in the acquired data [34]. To analyze the effects of the kernel with respect to the model- and noise-related errors, we simulated reconstruction algorithms depending on the presence of noise. Then, the nRMSE values calculated from the reference and the reconstructed images could be analyzed to evaluate the errors. More specifically, the nRMSE was calculated using the complex-valued reference and reconstructed images, as described in Equation (5), because simple subtraction from the sum of the squared reference and the reconstructed image could cancel out the residual aliasing artifacts in the difference images [35]. Based on the complex-valued difference calculation, the experiment using the noise-free simulation dataset demonstrated the effects of the reconstruction algorithm regarding the model error, while the experiments using the simulation dataset with an addition of 30 dB noise, the acquired phantom data and the acquired in vivo dataset demonstrated the efficiency of the reconstruction algorithm in the reduction of the combined errors due to the model- and the noise-related effects. According to our experiment with the noise-free simulation dataset, the severity of model-related error increased in the order of BK-2D-GRAPPA, EX-2D-GRAPPA, SK-2D-GRAPPA and LK-2D-GRAPPA. For the combined error caused by both the model and noise, the severity increased from the lowest to highest also in the order of BK-2D-GRAPPA, EX-2D-GRAPPA, SK-2D-GRAPPA and LK-2D-GRAPPA, as the calculated nRMSE values demonstrated.

Aside from GRAPPA, it should be noted that many advanced reconstruction methods are available for parallel imaging, as mentioned in Introduction. While some of the reconstruction methods, such as PRUNO, SAKE and LORAK are mostly used in 2D MRI due to the memory issue, HICU can be applied in 3D MRI using a convolutional framework. In addition, recent advances in deep-learning-based reconstruction methods show possible alternatives for parallel imaging [18,19,20]. With respect to analysis in kernel types for GRAPPA, the comparison of different size and shape of convolution kernels in parallel MRI reconstruction has been provided in a recent paper [24] but the study was focused on 2D imaging. In this study, we focused on the 2D-GRAPPA algorithm because it is one of the most widely used approaches for 3D imaging due to its straightforward nature. Thus, we analyzed kernels for 2D-GRAPPA, which is targeted for 3D data accelerated along the two PE directions and developed a new kernel.

While GRAPPA is straightforward to implement, the main limitation of GRAPPA is that the maximum acceleration factor is restricted by coil geometry. Thus, as demonstrated by our study, a larger kernel size does not always guarantee high image quality. When the acceleration factor was increased up to eight, substantial residual artifacts could be observed even after extending the kernel along the *k_x_* direction. In that respect, coil geometry and acceleration factors should be carefully chosen for best performance.

Although this study focused on providing experimental results with different kernels for 2D-GRAPPA, the different types of kernels can be applied to different convolutional reconstruction methods (e.g., SPIRit, SAKE, LORAKS, etc.) [24]. As a future work, boomerang-shaped kernels will be investigated for different reconstruction techniques. In addition, the proposed BK-2D-GRAPPA can be used in combination with recently proposed image-based deep learning reconstruction methods. As image-based learning requires baseline images as a training dataset, the improved quality of images reconstructed from undersampled datasets using the proposed BK-2D-GRAPPA can contribute to further improvement in reconstruction performance of the deep-learning networks.

## 6. Conclusions

In this paper, we proposed a boomerang-shaped kernel for the 2D GRAPPA algorithm based on theoretical and systematic analysis of the existing kernels to reconstruct 3D data that is undersampled in two dimensions. The proposed 2D GRAPPA algorithm with a boomerang kernel, called the BK-2D-GRAPPA algorithm, was compared with the various 2D GRAPPA algorithms using different shapes of kernels and showed that the BK-2D-GRAPPA algorithm reconstructed the best quality images from the accelerated k-space data, exhibiting the smallest differences from the reference image. In particular, the 3D boomerang kernel was able to provide fewer aliasing artifacts at a high AF. The number of memory allocations in the BK-2D-GRAPPA algorithm was also the second smallest among the 2D-GRAPPA algorithms. Thus, it is suggested to use the 3D boomerang kernel to reconstruct accelerated 3D MRI data along the two PE directions.

## Figures and Tables

**Figure 1 sensors-23-00093-f001:**
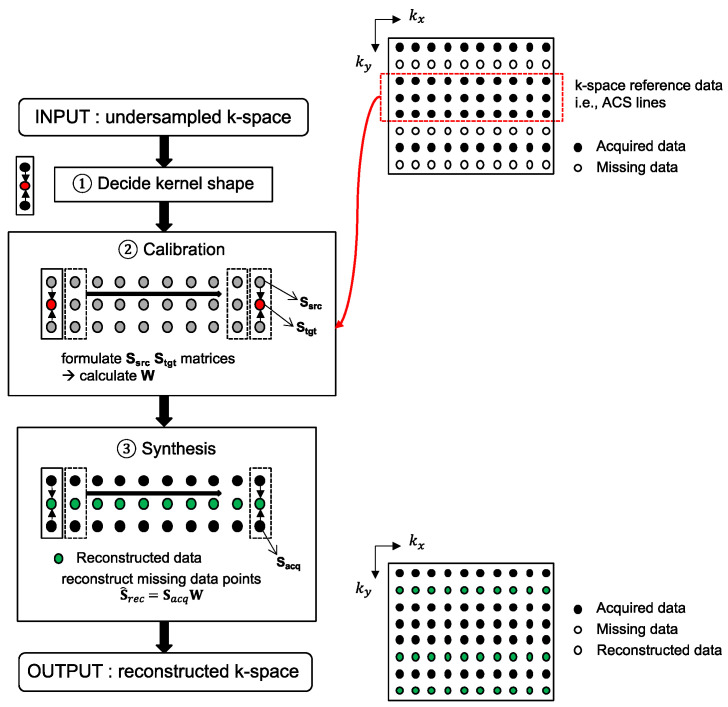
A flow chart of the 2D-GRAPPA reconstruction algorithm. To aid with explanation, *k_x_*-*k_y_* plane is displayed as an input. After determining the form of the kernel (e.g., estimating a missing data point using two adjacent obtained points along the *k_y_* direction), calibration is performed on the k-space reference lines. The precalculated weighting values are then subsequently used for reconstruction of non-ACS line regions.

**Figure 2 sensors-23-00093-f002:**
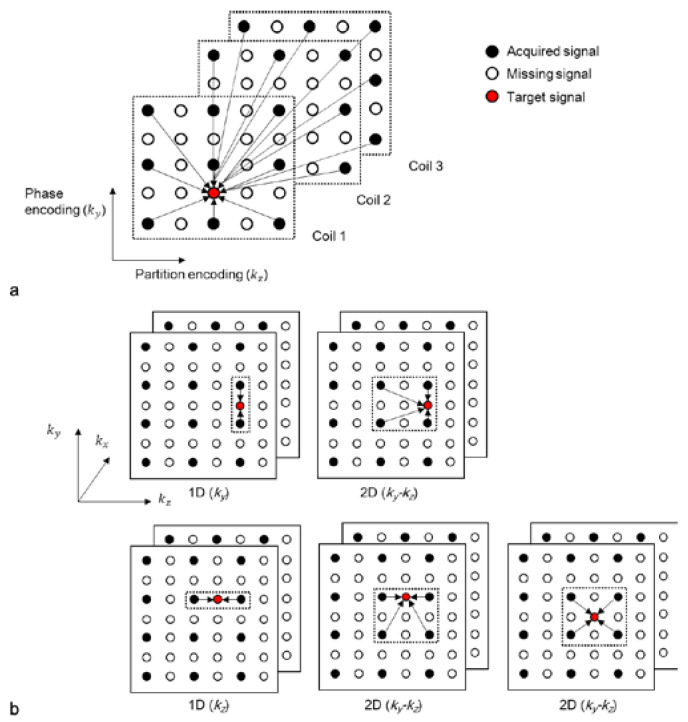
(**a**) A schematic diagram of the 2D-GRAPPA algorithm and (**b**) different types of kernels. Each circle represents a data point in the k-space of each coil. A missing data point in k-space can be estimated by a linear combination of acquired signals.

**Figure 3 sensors-23-00093-f003:**
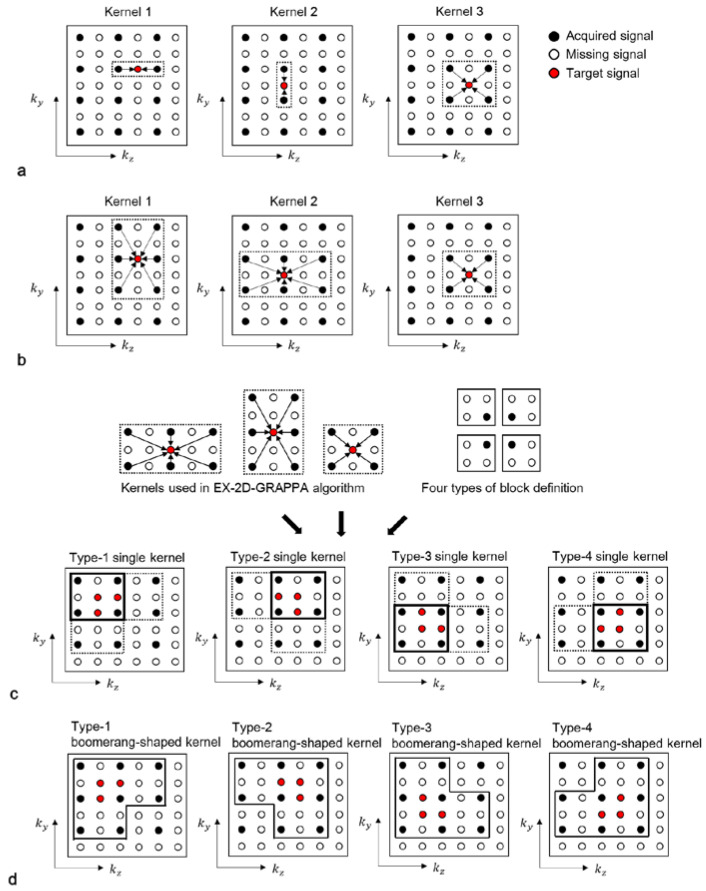
Schematic diagrams of LK-2D-GRAPPA, EX-2D-GRAPPA, SK-2D-GRAPPA and BK-2D-GRAPPA algorithms with basis kernels. (**a**) The basis kernels of the LK-2D-GRAPPA algorithm are two 1D kernels and a 2D kernel, which use acquired data points in each PE direction and *k_y_*-*k_z_* plane, respectively. (**b**) The EX-2D-GRAPPA algorithm utilizes three types of 2D kernels; two of them are extended from the LK-2D-GRAPPA algorithm. (**c**) A single square kernel is created with the common acquired data points utilized with the kernels of the EX-2D-GRAPPA algorithm. (**d**) A boomerang kernel is produced with all the acquired data points used with the kernels of the EX-2D-GRAPPA algorithm.

**Figure 4 sensors-23-00093-f004:**
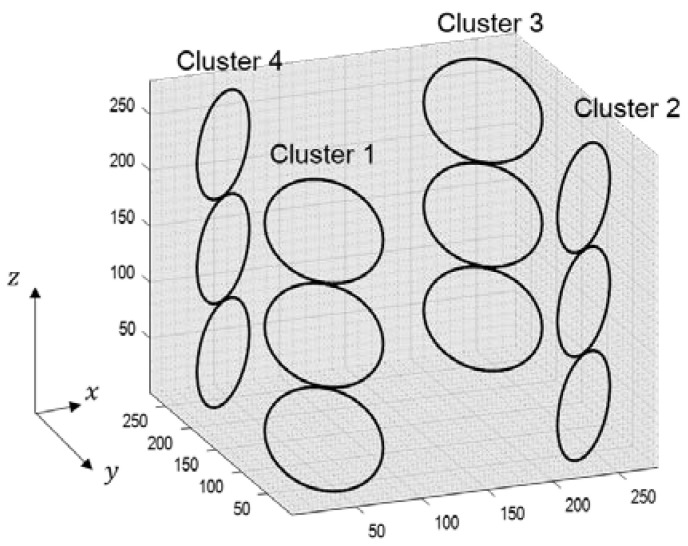
The structure of 12-channel head coil for computer simulation.

**Figure 5 sensors-23-00093-f005:**
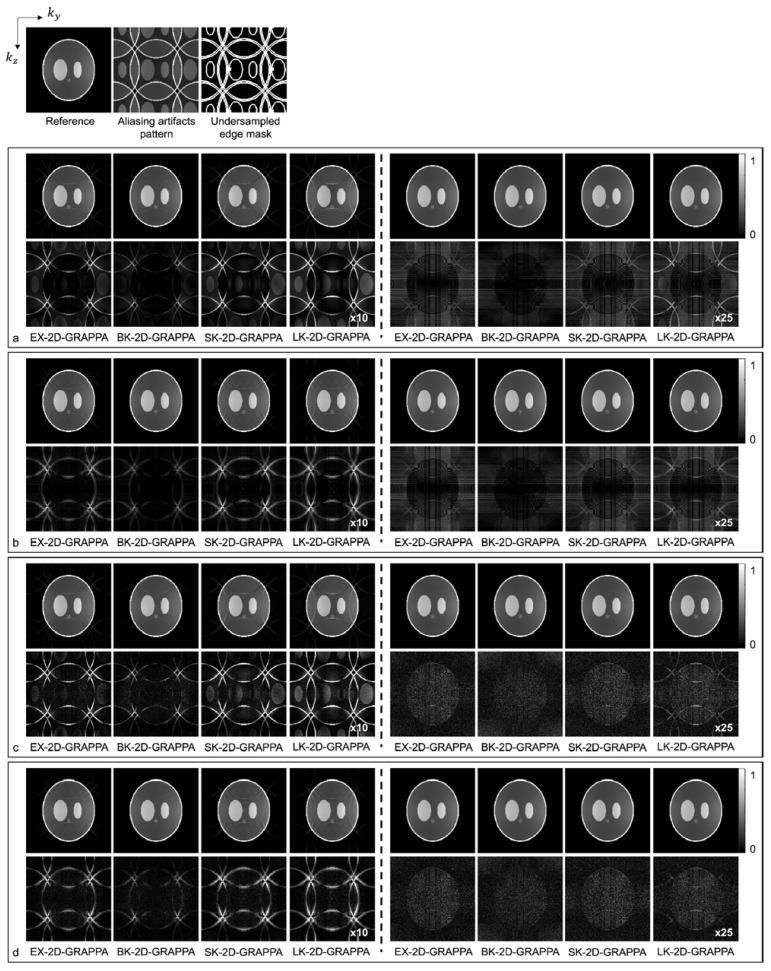
The images reconstructed by different 2D-GRAPPA algorithms from (**a**,**b**) noise-free and (**c**,**d**) 30 db noise-added simulation data with AF = 4 (2 × 2). The 24 × 24 k-space reference lines were (**a**,**c**) excluded and (**b**,**d**) included for reconstruction. Columns 1–4: basis kernel. Columns 5–8: expanded kernel.

**Figure 6 sensors-23-00093-f006:**
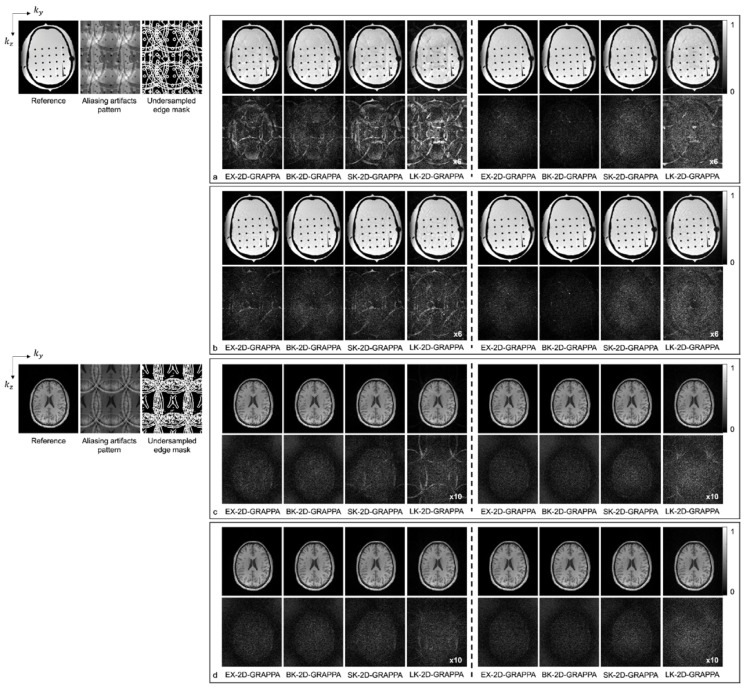
The reconstructed images of acquired anthropomorphic skull phantom (**a**,**b**) and human brain (**c**,**d**) data with 24 × 24 k-space reference lines and AF = 4 (2 × 2). The k-space reference lines were (**a**,**c**) excluded and (**b**,**d**) included at the end of the reconstruction. Columns 1–4: basis kernel. Columns 5–8: expanded kernel.

**Figure 7 sensors-23-00093-f007:**
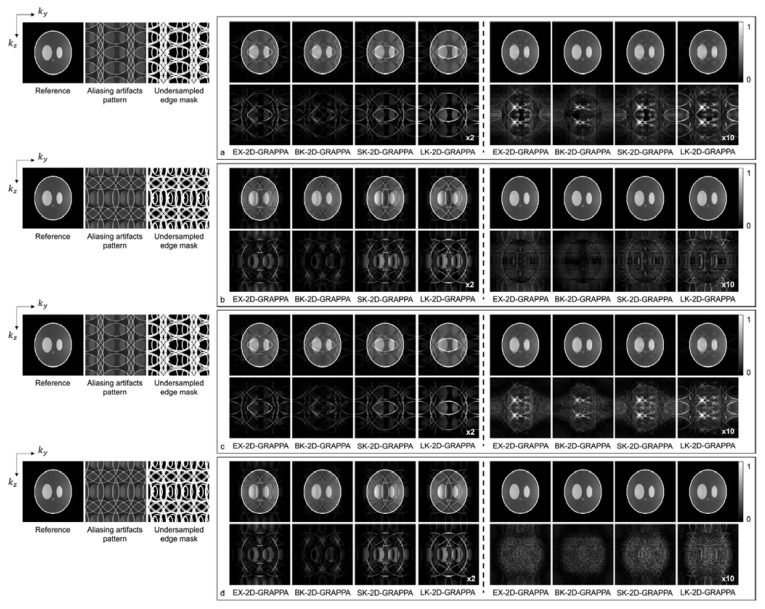
The reconstruction results of 2D-GRAPPA algorithms for simulation data with AF = 8. (**a**–**d**) were reconstructed from noise-free and noise-added data, respectively. Images in (**a**,**c**) were obtained when *k_z_* direction was more accelerated (2 × 4), and images in (**b**,**d**) were obtained when *k_y_* direction was more accelerated (4 × 2). Columns 1–4: basis kernel. Columns 5–8: expanded kernel.

**Figure 8 sensors-23-00093-f008:**
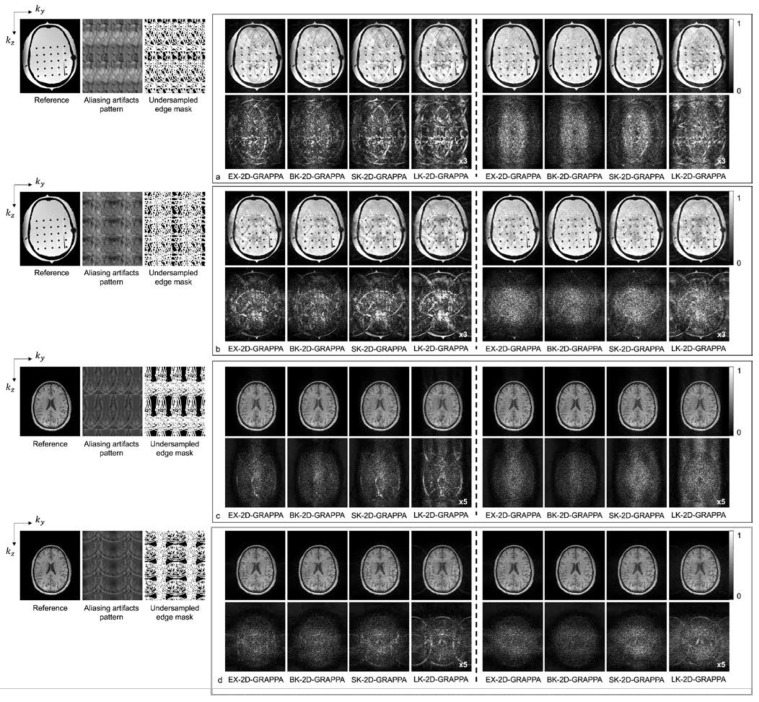
The reconstruction results of 2D-GRAPPA algorithms for (**a**,**b**) anthropomorphic head phantom and (**c**,**d**) in vivo data with AF = 8. The top rows in (**a**,**c**) were obtained when *k_z_* direction was more accelerated (2 × 4) and those in (**b**,**d**) were obtained when *k_y_* direction was more accelerated (4 × 2). Columns 1–4: basis kernel. Columns 5–8: expanded kernel.

**Figure 9 sensors-23-00093-f009:**
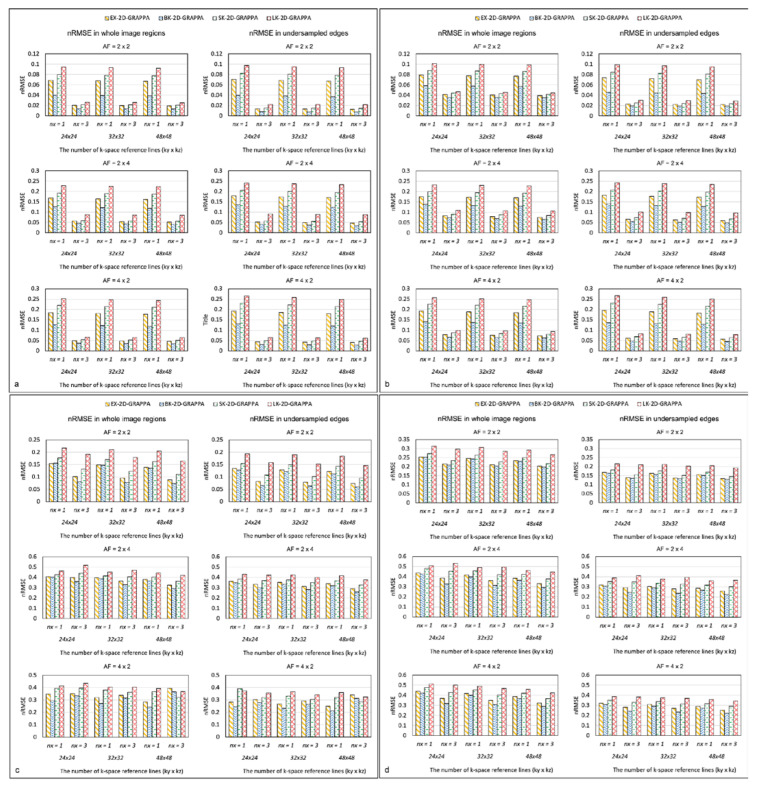
The nRMSE of 2D-GRAPPA algorithms in (**a**) noise-free, (**b**) noise-added, (**c**) phantom and (**d**) in vivo data. BK-2D-GRAPPA algorithm shows the best performance. *nx* represents the kernel size along the x direction. *nx* = 1: basis kernel, *nx* = 3: expanded kernel.

**Table 1 sensors-23-00093-t001:** The number of memory allocations and arithmetic operations in each algorithm.

# of Target and Source Data		Calibration Step		Synthesis Step	
		Matrix (S_src_,S_tgt_) Formulation Number	Arithmetic Operation Number	Matrix (S_acq_) Formation Number	Arithmetic Operation Number
Target: 1, source: *n_src_*		3*N_f_n_src_J* + *N_f_J*	(*n_src_J*)^2^(2*N_f_* − 1) + *n_src_J^2^*(2*N_f_* + 2*n_src_J* − 2)	*N_rec_n_src_J* + *n_src_J^2^*	*N_rec_J*(2*n_src_J* − 1)
**Algorithms**	**# of target and source data**	**Calibration step**		**Synthesis step**	
		Matrix (**S_src_**,**S_tgt_**) formulation number	Arithmetic operation number	Matrix (**S_acq_**) formation number	Arithmetic operation number
LK-2D-GRAPPA	Kernel 1—target: 1, source: 2 Kernel 2—target: 1, source: 2 Kernel 3—target: 1, source: 4	(3 × 12^2^ × 2 + 12^2^) + (3 × 12^2^ × 2 + 12^2^) + (3 × 12^2^ × 4 + 12^2^) = 3688	{(2 × 12)^2^23 + (2 × 12^2^)70} + {(2 × 12)^2^23 + (2 × 12^2^)70} + {(4 × 12)^2^23 + (4 × 12^2^)118} = 187,776	(250,000 × 2 × 12^2^ + 2 × 12^2^) + (250,000 × 2 × 12^2^ + 2 × 12^2^) + (250,000 × 4 × 12^2^ + 4 × 12^2^) = 24,001,152	250,000 × 12 × 47 + 250,000 × 12 × 47 + 250,000 × 12 × 95 = 567,000,000
EX-2D-GRAPPA	Kernel 1—target: 1, source: 6 Kernel 2—target: 1, source: 6 Kernel 3—target: 1, source: 4	(3 × 12^2^ × 6 + 12^2^) + (3 × 12^2^ × 6 + 12^2^) + (3 × 12^2^ × 4 + 12^2^) = 7344	{(6 × 12)^2^23 + (6 × 12^2^)70} + {(6 × 12)^2^23 + (6 × 12^2^)70} + {(4 × 12)^2^23 + (4 × 12^2^)118} = 480,384	(250,000 × 6 × 12^2^ + 6 × 12^2^) + (250,000 × 6 × 12^2^ + 6 × 12^2^) + (250,000 × 4 × 12^2^ + 4 × 12^2^) = 48,002,304	250,000 × 12 × 143 + 250,000 × 12 × 143 + 250,000 × 12 × 95 = 1,143,000,000
SK-2D-GRAPPA	Kernel 1—target: 3, source: 4	(3 × 12^2^ × 4 + 12^2^) = 1872	(4 × 12)^2^23 + (3 × 4 × 12^2^)118 = 256,896	(250,000 × 4 × 12^2^ + 4 × 12^2^) = 12,000,576	3 × 250,000 × 12 × 95 = 885,000,000
BK-2D-GRAPPA	Kernel 1—target: 3, source: 8	(3 × 12^2^ × 8 + 12^2^) = 3600	(8 × 12)^2^23 + (3 × 8 × 12^2^)214 = 951,552	(250,000 × 8 × 12^2^ + 8 × 12^2^) = 24,001,152	3 × 250,000 × 12 × 191 = 1,719,000,000

## Data Availability

Not applicable.

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
