# Peer review of "A 2D-GRAPPA Algorithm with a Boomerang Kernel for 3D MRI Data Accelerated along Two Phase-Encoding Directions"

_sensors, 2022, doi:10.3390/s23010093_

Round 1
Reviewer 1 Report
The submitted manuscript proposed a 2D-GRAPPA algorithm with boomerang kernel. The experimental results show that the images reconstructed using BK 2D GRAPPA algorithm have less artifacts compared to other 2D-GRAPPA algorithm. The authors also claim that the memory allocation in BK-2D-GRAPPA is the second smallest in 2D-GRAPPA algorithms.
However, GRAPPA is a well-developed algorithm in parallel imaging, and a lot of advanced methods have been proposed recently, such as PRUNO [1], SAKE [2], AC-LORAKs [3] and HICU [4]. All these methods rely on the Linear Predictability in the MRI data [5]. Although PRUNO, SAKE and LORAK are mostly used in 2D MRI due to the memory issue, HICU has been applied in 3D MRI successfully using convolutional framework. The comparison of different size and shape of convolution kernels in parallel MRI reconstruction is also studied in a recent paper [6]. Overall, I think the current manuscript is not ready for publication due to limited novelty and the missing comparison with the latest reconstruction methods.
[1] Zhang J, Liu C, Moseley ME. Parallel reconstruction using null operations. Magn Reson Med. 2011;66:1241- 1253.
[2] Shin PJ, Larson PE, Ohliger MA, et al. Calibrationless parallel im-aging reconstruction based on structured low- rank matrix comple-tion. Magn Reson Med. 2014;72:959- 970.
[3] Haldar JP. Low- rank modeling of local k- space neighborhoods (LORAKS) for constrained MRI. IEEE Trans Med Imaging. 2014;33:668- 681.
[4] S. Zhao, L. C. Potter, and R. Ahmad, “High-dimensional fast convolutional framework (HICU) for calibrationless MRI,” Magn. Reson. Med., vol. 86, pp. 1212–1225, 2021.
[5] Haldar JP, Setsompop K. Linear predictability in magnetic resonance imaging reconstruction: leveraging shift-invariant Fourier structure for faster and better imaging. IEEE Signal Process Mag. 2020; 37: 69- 82.
[6] Rodrigo A. Lobos,Justin P. Haldar; On the shape of convolution kernels in MRI reconstruction: Rectangles versus ellipsoids. Magn Reson Med. 2022;87:2989–2996
Reviewer 2 Report
The authors proposed a new approach to accelerate the 3D MRI data reconstruction. The results look promising, but I hope authors could address some of my doubts.
1. There are many typos and grammar errors in the writing. The manuscript is also not organized well. Please spend time to revise the manuscript.
2. The introduction is too short and not enough to explain the background.
3. Some figures and are not clear. Please change the resolution or modify the format.
Reviewer 3 Report
The study is good but needs more improvement.
1. The 2. Theory section should be moved to the Methods and Materials section, with its name changed from Theory to Methods or Methodologies.
2. Move Figure 1 to the end of "2.1. 2D-GRAPPA Algorithm and its Kernels"
3. Move Figure 2 to the end of "2.3. Extended Kernel (EX)-2D-GRAPPA Algorithm".
4. A brief explanation of the Methods and Materials section to clarify the working mechanism with a figure showing the workflow from inputs to outputs.
5. A conclusion section should be added.
6. What are the limitations you faced during work and future work?
7. A new section should be added after the introduction titled “Related Works” to analyze and interpret the techniques and results of at least 15 previous studies.
Round 2
Reviewer 1 Report
Thanks for addressing my comments.
Reviewer 2 Report
The revision has improved the quality of the manuscript
Reviewer 3 Report
Thanks for the authors' response and answers to all questions